# Real sounds influence postural stability in people with vestibular loss but not in healthy controls

Anat V. Lubetzky[1]*, Maura Cosetti[2,3], Daphna Harel[4], Marlee Sherrod[4], Zhu Wang[5], Agnieszka Roginska[6], Jennifer Kelly[2,3]

1 Department of Physical Therapy, Steinhardt School of Culture, Education and Human Development, New York University, New York, NY, United States of America, 2 Department of Otolaryngology-Head and Neck Surgery, Icahn School of Medicine at Mount Sinai, New York, NY, United States of America, 3 Ear Institute, New York Eye and Ear Infirmary of Mount Sinai, New York, NY, United States of America, 4 Department of Applied Statistics, Social Science and Humanities, Steinhardt School of Culture Education and Human Development, New York University, New York, NY, United States of America, 5 New York University, Computer Science Department, Courant Institute of Mathematical Sciences, New York, NY, United States of America, 6 Department of Music and Performing Arts Professions, Steinhardt School of Culture, Education and Human Development, New York University, New York, NY, United States of America

* anat@nyu.edu

**Data Availability Statement:** The dataset used in this manuscript can be found at: Lubetzky, Anat (2024), "Real Sounds Influence Postural Stability in People with Vestibular Loss but Not in Healthy

## Abstract

### Objective

What we hear may influence postural control, particularly in people with vestibular hypofunction. Would hearing a moving subway destabilize people similarly to seeing the train move? We investigated how people with unilateral vestibular hypofunction and healthy controls incorporated broadband and real-recorded sounds with visual load for balance in an immersive contextual scene.

### Design

Participants stood on foam placed on a force-platform, wore the HTC Vive headset, and observed an immersive subway environment. Each 60-second condition repeated twice: static or dynamic visual with no sound or static white noise or real recorded subway station sounds [real] played from headphones.

### Setting

Human motion laboratory.

### Participants

41 healthy controls (mean age 52 years, range 22–78) and 28 participants with unilateral peripheral vestibular hypofunction (mean age 61.5, 27–82)

### Main outcome measures

We collected center-of-pressure (COP, anterior-posterior, medio-lateral) from the force-platform and head (anterior-posterior, medio-lateral, pitch, yaw, roll) from the headset and quantified root mean square velocity (cm/s or rad/s)

Controls", Mendeley Data, V1, doi: 10.17632/
cwy56jmwwt.1.

**Funding:** This study was funded by an
R21DC018101 Early Career Researcher grant from
the National Institute on Deafness and Other
Communication Disorders (NIDCD). The sponsors
had no role in the study design, collection, analysis
and interpretation of data; in the writing of the
manuscript; or in the decision to submit the
manuscript for publication. The content is solely
the responsibility of the authors and does not
necessarily represent the official views of the
National Institutes of Health. This work was also
supported in part through the computational and
data resources and staff expertise provided by
Scientific Computing and Data at the Icahn School
of Medicine at Mount Sinai and supported by the
Clinical and Translational Science Award (CTSA)
grant UL1TR004419 from the National Center for
Advancing Translational Science.

**Competing interests:** The authors have declared
that no competing interests exist.

## Results

Adjusting for age, the vestibular group showed significantly more sway than controls on:
COP medio-lateral (no sound or real with static or dynamic visual); COP anterior-posterior
(only on dynamic visuals in the presence of either sound); head medio-lateral and anterior-
posterior (all conditions), head pitch and yaw (only on dynamic visuals in the presence of
either sound). A significant increase in sway with sounds was observed for the vestibular
group only on dynamic visuals COP anterior-posterior and head yaw (real) and head ante-
rior-posterior and pitch (either sound).

## Conclusions

The addition of auditory stimuli, particularly contextually-accurate sounds, to a challenging,
standing balance task in real-life simulation increased sway in people with vestibular hypo-
function but not in healthy controls.

## Trial registration

### Clinical trial registration

This study was registered on clinicaltrials.gov at the following link: https://clinicaltrials.gov/
study/NCT04479761.

## Introduction

Maintaining balance while standing, moving, or walking requires continuous modifications to
the proportion that each sensory system contributes to overall stability given the environmen-
tal conditions [1–4]. When input from one sensory system becomes unreliable or destabilizing,
such as a moving subway train or a flow of people in a busy street, a healthy postural control
system reweights that information (i.e., reduces its contribution) and relies on other sensory
cues to maintain balance. Individuals with known sensory impairment, such as vestibular
hypofunction, rely heavily on visual cues and may lose their balance when these cues are com-
plex [5,6]. Typically visual, vestibular, and somatosensory system are considered as sensory
inputs for postural control yet recent studies suggest that auditory cues may be integrated as
well, particularly in those with balance deficits, such vestibular hypofunction [7–9]. One pro-
posed theory regarding the integration of sound into postural control is that of the "auditory
anchor" [8,10,11]. According to this theory, stationary sound sources provide information that
helps the brain structure a spatial map of the environment, which the brain uses for stabiliza-
tion [12]. For example, instability induced by standing on a soft surface (considered to reduce
somatosensory input) was improved in the presence of auditory cues [13]. The auditory
anchor theory also suggests that if the sound source is suddenly changed [14] or moves, e.g.,
'jumps' from one side to the other [15], it will interfere with the spatial hearing map and lead
to destabilization (i.e., increased postural sway).

Do auditory cues function similarly to visual cues for balance? Would hearing a moving
subway destabilize people with vestibular hypofunction similarly to seeing the train's move-
ment? The few studies that implemented natural sounds sources (e.g., chatter, fountain etc.) in
the study of balance provide general support for the role of audition in postural control but
offer inconclusive evidence regarding what that involvement may look like. Gandemer et al.

[16] found that healthy adults decreased sway when ecologically valid sound sources were added (from 3 to 10 isolated sources, such as a fountain sound, motor sound etc.). The decrease in sway was larger in a rich auditory immersive environment compared with an isolated source. Guigo et al. [17] found that 10 individuals with bilateral vestibular loss and normal hearing increased their sway when standing with eyes closed and listening to rotating cocktail party noise (compared with silence) via headphones.

The role of sound in postural control has been primarily studied within tasks that include complete elimination of visual input, e.g., standing with eyes closed [13,18]. Context has been shown to have an important impact on balance performance, potentially induced by cognitive and emotional aspects, such as postural threats or fear of imbalance or symptoms related to past experiences within specific environments [19]. Sensory loss (such as vestibular hypofunction) may also explain context-dependent postural differences in response to visual stimuli, as well as compensatory strategy and reserves for postural control. In order to understand how sounds may be used for balance in daily living, there is a need to investigate how postural control changes in response to broadband noise and real-life contextual sounds, especially when these sounds are combined with dynamic visual environments similar to real life for sighted individuals. Beyond healthy controls, it is imperative to study balance behavior in individuals with a clearly defined preexisting condition that may affect multisensory integration for postural control, such as vestibular hypofunction, both because their behavior may differ from that of healthy controls, and also because the clinical implications may directly transfer to vestibular rehabilitation. Finally, because people with hearing loss have been shown to respond less to auditory cues than those with normal hearing [13], it is important to study the role of sound in postural control in patients with a primary complaint of vestibular loss and normal hearing.

This study aimed to understand the impact of sounds on postural control in people with vestibular hypofunction vs. healthy controls, all with normal hearing. We tested the role of generated broadband noise and real recorded sounds in postural control in an ecologically valid immersive subway environment. We expected that both groups would show a significant increase in sway with increased visual complexity, which would be larger in the vestibular group. We hypothesized that real recorded sounds would increase sway, mostly in the vestibular group, of smaller magnitude than changes in sway observed with visual load [13]. Because static (stationary) white noise (broadband) has been shown to serve as an "auditory anchor" [13,17] (helping to reduce postural sway) we expected the white noise to be associated with a reduction in postural sway.

## Methods

The study was approved by the Biomedical Research Alliance of New York LLC Institutional Review Board (BRANY IRB, study #20-02-278-05) and registered on clinicaltrials.org (NCT04479761). Recruitment began on September 15th, 2021, and ended on December 23rd, 2023.

### Eligibility criteria

Participants with unilateral peripheral vestibular hypofunction were recruited from vestibular rehabilitation at the New York Eye and Ear Infirmary of Mount Sinai. They presented with: 1) a complaint of head motion provoked instability or dizziness affecting their functional mobility and quality of life; 2) at least 1 positive finding indicating unilateral vestibular hypofunction on the following clinical tests: head thrust, subjective visual vertical and horizontal, post head shaking nystagmus, spontaneous and gaze holding nystagmus [20] and 3) A score of at least 16

(mild handicap) on the Dizziness Handicap Inventory (DHI) [21]. Participants were excluded from the vestibular group for any unstable peripheral lesion, e.g., Meniere's Disease, Perilymphatic Fistula, Superior Canal Dehiscence, or Acoustic Neuroma. Healthy controls were recruited from the university and hospital community. Healthy controls were excluded for vestibular symptoms (DHI≥16) and / or history of vestibular rehabilitation. All participants had normal Hearing defined as an unaided PTA < 26dB HL (0.5–4 kHz) bilaterally. Individuals above 65 years of age with symmetric high frequency loss leading to an unaided PTA < 40 dB (0.5-4KHz) were included as well [22]. Exclusion criteria for both groups included a medical diagnosis of peripheral neuropathy; lack of protective sensation based on the Semmes-Weinstein 5.07 Monofilament Test; [23] visual impairment above 20/63 (NYS Department of Motor Vehicle cutoff for driving) on the Early Treatment Diabetic Retinopathy Study (ETDRS) Acuity Test that cannot be corrected with lenses; conductive hearing loss; pregnancy; any neurological condition interfering with balance or walking (e.g. multiple sclerosis, Parkinson's disease, stroke); acute musculoskeletal pain at time of testing; currently seeking medical care for another orthopaedic condition; and inability to read an informed consent in English or Spanish.

## System

We developed the system in C# with Unity 2019.4.16f1(64-bit) (©Unity Technologies, San Francisco, CA, USA). The subway scene was a 3D subway station model with two visual levels. The dynamic visual level utilized a crowd generation module to create avatars walking in groups on the platform and staircase. Using a collision avoidance algorithm, the avatars avoided colliding with the participant or other avatars. Subway trains were active on all the 3 subway rails. A subway car passed by the participant on each rail every 16 seconds. The static visual level rendered everything in the dynamic visual level but paused the motions of avatars and subway trains; thus, there were stationary avatars and subway trains around the participants. In addition, we integrated temporary anti-aliasing (TAA) in the subway scene to reduce the aliasing caused by the jagged edges. See S1 and S2 Videos.

Each visual level was combined with 3 possible auditory levels: none, white noise, and real.

In the condition referred to as "none," the participants were wearing noise cancelling headphones with no sounds. For the white noise condition, s static white noise was generated at 44.1kHz sampling rate by independent and identically distributed random variables. The white noise was played constantly over the entire 60s trial with a spatialized audio source at the eye level height and 1.63 meter away in front of the participant. The real recorded sounds included ambient sounds, including the station's noise, people chatting, announcements recorded from a real subway station; footsteps, and the trains passing by. The sounds of the trains were synced with the visual of the train's movement where applicable.

## Procedures

All participants signed a written informed consent prior to commencing the study procedures. The study included a screening session and a postural control testing session. The screening session included the following: a written informed consent, visual and monofilament screen of eligibility criteria, comprehensive behavioral audiometry, the caloric portion of the Videonystagmography (VNG) test and Video Head Impulse Test (vHIT). Participants also completed a demographic questionnaire as well the Activities Specific Balance Confidence Scale and the Dizziness Handicap Inventory scale [24].

The participants wore the HTC Vive Pro headset (HTC Corporation, Taoyuan City, Taiwan) and a cable-connected Bose QuietComfort® 35 II around-ear headphones (Bose

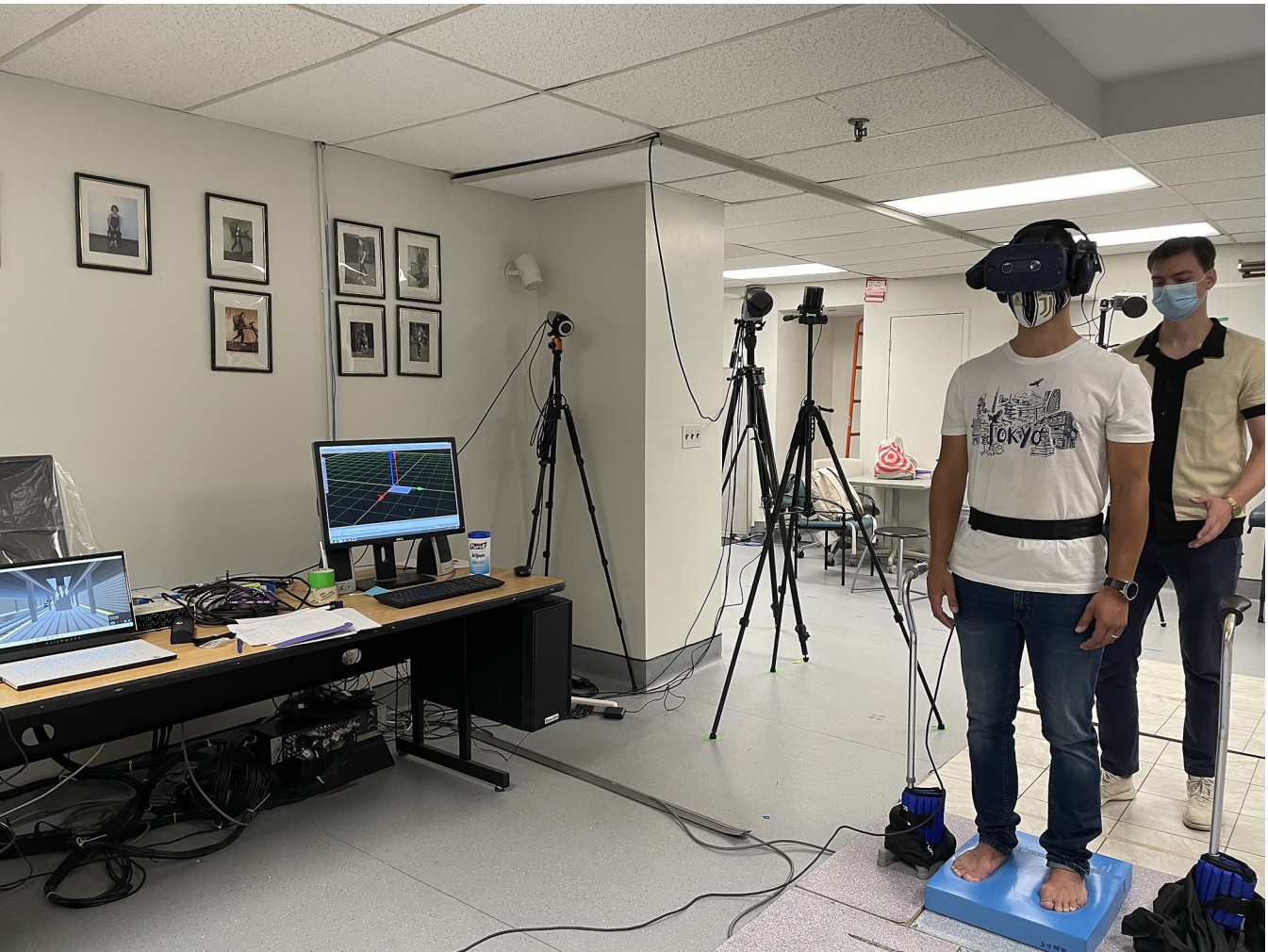

**Fig 1. Experimental setup.**

Corporation, Framingham, MA). They stood with their feet hips-width apart on a foam surface (AIREX, Sins, Switzerland) placed on a force platform (Kistler, Winterthur, Switzerland). All conditions were randomized. Feet position was marked on the foam to maintain consistency of base of support between conditions. See Fig 1 for the experimental setup. Sounds were played at the highest level that was comfortable to the participant. Scenes were 60 seconds long. There were 6 combinations: 2 visuals (static, dynamic) for each of 3 sounds (none, white noise, real), each repeated twice. Participants completed the Simulator Sickness Questionnaire (SSQ), a 15-items questionnaire, before, mid and at the end of testing. They rated their overall symptoms as none, slight, moderate or severe per item [16,17,25,26].

## Data processing and outcome measures

Head sway was recorded at 90Hz via custom-made software programmed in Unity. Center-of-pressure (COP) data was recorded at 100Hz. Data processing was performed in Matlab R2024a (Mathworks, Natick, MA). A low-pass 4[th] order Butterworth filter was applied with 10 Hz as the cutoff frequency [27]. The first 5 seconds of each 60 second trial were removed, and the remaining 55 seconds were used to quantify COP and head Root Mean Square Velocity

(RMSV, in cm/s) in the anteroposterior and mediolateral directions, as well as Head RSMV Pitch, Yaw, and Roll (rad/s). RMSV is defined the difference in position between two data points divided by the average time interval, with the velocities at each point squared then summed. The square root of this sum is then divided by the number of data points. Higher values reflect more postural sway.

### Data analysis

We described demographic information for each group separately (mean, standard deviation, median, minimum, maximum or count, % as applicable) and compared between groups using the appropriate statistical test.

To evaluate the main effect of group, we fit a linear mixed effects model for each outcome measure (COP VRMS AP and ML, Head VRMS AP, ML, Pitch, Yaw, Roll) using a log transformation of the outcomes. The models were fit separately for each auditory condition (none, white noise, real). Each model included the main effect of group (control, vestibular) and visual condition (static, dynamic) as well as their two-way interaction. Next, to evaluate the main effect of sound, we fit the same models, now including the 3 auditory levels per group and visual condition. All models adjusted for age. Although these models are equivalent to a 3-way interaction of group by visual by auditory condition, we chose this modeling approach to simplify interpretation. These models maximize the information we can obtain from the data by accounting for the inherent multi-level study design (person, conditions, repetitions). Since each person completes various trials for each condition, the linear mixed effects model accounts for these sources of variability [28]. *P*-values for the fixed effects were calculated using the Satterthwaite approximation for the degrees of freedom for the T-distribution [29]. The Model Estimated difference (MED) is reported as an indication of effect size.

## Results

### Sample

A detailed description of our sample can be seen in Table 1. The mean age of the control group was 52 (range 22 to 78) and of the vestibular group 61.5 (range 27 to 82).

### COP

**ML.**   Adjusting for age, the vestibular group was significantly higher than controls on static visual with real sounds (P = 0.04, MED = 0.22) and dynamic visual no sounds (P = 0.04, MED = 0.25). A significant main effect of visual was observed for both groups across sound conditions (none, static, real, P<0.001) with no interactions. No significant main effect of sound was observed. See Fig 2 for visualization of the model and S1 File for all model estimates.

**AP.**   Adjusting for age the vestibular group was significantly higher than controls only on dynamic visuals with either sound (white noise P = 0.03, MED = 0.48, real P = 0.02, MED = 0.53). A significant main effect of visual was observed for both groups across sound conditions (none, white noise, real, P<0.001, see Fig 3 for visualization of the model and S1 File for all model estimates) with no interactions. A significant increase in sway with real sounds vs. no sound was observed for the vestibular group on dynamic visuals (P = 0.04, MED = 0.23).

### HEAD

**ML.**   Adjusting for age, the vestibular group was significantly higher than controls on all conditions (static visual: no sound P = 0.008, MED = 0.16; white noise P = 0.034, MED = 0.13; real P = 0.005, MED = 0.19; dynamic visual: no sound P = 0.018, MED = 0.17; white noise

**Table 1. Descriptive statistics.**

| | Control (N = 41) | Vestibular loss (N = 28) | Significance |
|---|---|---|---|
| **Age** | | | P = 0.03[T] |
| Mean (SD) | 52.0 (17.9) | 61.5 (14.6) | |
| Median [Min, Max] | 55.0 [22.0, 78.0] | 64.0 [27.0, 82.0] | |
| **Sex** | | | P = 0.69[Z] |
| Female | 20 (48.8%) | 15 (53.6%) | |
| Male | 21 (51.2%) | 13 (46.4%) | |
| **Weight in kg** | | | P = 0.58[T] |
| Mean (SD) | 71.7 (15.0) | 72.8 (13.1) | |
| Median [Min, Max] | 72.6 [47.2, 103] | 72.8 [49.9, 102] | |
| **Height cm** | | | P = 0.33[T] |
| Mean (SD) | 171 (11.7) | 168 (11.5) | |
| Median [Min, Max] | 170 [150, 198] | 170 [140, 188] | |
| **Race** | | | P = 0.37[C] |
| Asian / Pacific Islander | 3 (7.3%) | 0 (0%) | |
| Asian American | 4 (9.8%) | 4 (14.3%) | |
| African American | 2 (4.9%) | 0 (0%) | |
| White | 24 (58.5%) | 18 (64.3%) | |
| Spanish / Hispanic | 4 (9.8%) | 4 (14.3%) | |
| Other | 4 (9.8%) | 2 (7.1%) | |
| **Comorbidities** | | | |
| Hypertension | 6 (14.6%) | 7 (25.0%) | P = 0.28[Z] |
| Diabetes | 1 (2.4%) | 7 (25.0%) | P = 0.004[Z] |
| Migraines | 5 (12.2%) | 4 (14.3%) | P = 0.80[Z] |
| Cardiovascular Disease | 0 (0%) | 4 (14.3%) | P = 0.005[Z] |
| Anxiety/ depression | 9 (22.0%) | 4 (14.3%) | P = 0.80[Z] |
| **Falls in the past year** | | | P = 0.037[M] |
| Mean (SD) | 0.122 (0.510) | 0.536 (1.14) | |
| Median [Min, Max] | 0 [0, 3.00] | 0 [0, 5.00] | |
| **Do you currently exercise?** | | | P = 0.08[Z] |
| Yes | 38 (92.7%) | 22 (78.6%) | |
| **ABC (%)** | | | P<0.001[T] |
| Mean (SD) | 96.4 (5.12) | 74.1 (18.8) | |
| Median [Min, Max] | 98.8 [74.4, 100] | 75.9 [8.13, 99.4] | |
| **Total DHI Score** | | | P<0.001[T] |
| Mean (SD) | 0 (0) | 35.6 (18.1) | |
| Median [Min, Max] | 0 [0, 0] | 36.0 [4.00, 74.0] | |
| Missing | 0 (0%) | 1 (3.6%) | |
| **Chronicity of vestibular symptoms (years)** | | | |
| Mean (SD) | NA | 2.46 (2.82) | |
| Median [Min, Max] | NA | 2 (0.125, 12) | |
| **SSQ baseline** | | | P<0.001[T] |
| Mean (SD) | 0.341 (0.617) | 2.57 (2.47) | |
| Median [Min, Max] | 0 [0, 2.00] | 2.00 [0, 9.00] | |
| Missing | 0 (0%) | 0 (0%) | |
| **SSQ post** | | | P = 0.006[T] |
| Mean (SD) | 0.415 (1.16) | 3.37 (5.12) | |

*(Continued)*

**Table 1.** (Continued)

|  | Control (N = 41) | Vestibular loss (N = 28) | Significance |
|---|---|---|---|
| Median [Min, Max] | 0 [0, 6.00] | 2.00 [0, 22.0] |  |
| Missing | 0 (0%) | 1 (3.6%) |  |
| **Unilateral Caloric Weakness (asymmetry > 25%)** |  |  | P<0.001[Z] |
| Yes | 5 (12.2%) | 25 (89.3%) |  |
| Missing | 0 (0%) | 0 (0%) |  |
| **Lateral vHIT gain < 0.7** |  |  | P<0.001[Z] |
| Yes | 0 (0%) | 12 (42.9%) |  |
| Missing | 1 (2.4%) | 5 (17.9%) |  |
| **PTA worse hearing ear (dB)** |  |  | P = 0.002[T] |
| Mean (SD) | 14.2 (8.49) | 22.0 (10.4) |  |
| Median [Min, Max] | 12.5 [1.25, 33.8] | 18.1 [5.00, 45.0] |  |
| **Word Discrimination Score (%)** |  |  | P = 0.06[T] |
| Mean (SD) | 98.3 (3.22) | 96.9 (3.30) |  |
| Median [Min, Max] | 100 [88.0, 100] | 96.0 [92.0, 100] |  |

[T] T test; [Z] Z test of proportions; [C] Chi-Square test; [M] Mann-Whitney test.

P = 0.006, MED = 0.2; real P = 0.006, MED = 0.22). A significant main effect of visual was observed for both groups across sound conditions (none, static, real, P<0.001) with no interactions. No significant main effect of sound was observed. See Fig 4 and S1 File.

**AP.**   Adjusting for age, the vestibular group was significantly higher than controls on all conditions (static visual: no sound P = 0.01, MED = 0.21; white noise P = 0.004, MED = 0.27;

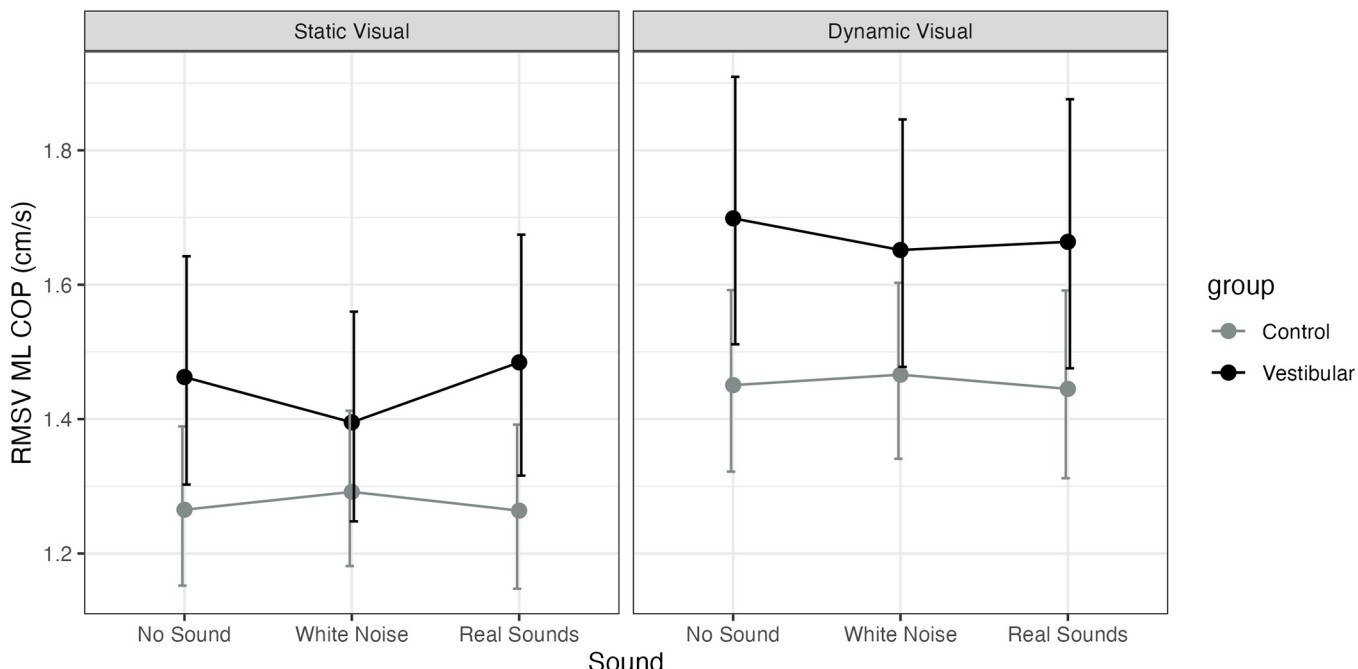

**Fig 2. Model estimated marginal means for medio-lateral center-of-pressure (COP) Root Mean Square Velocity (RMSV, cm/s) and their 95% confidence intervals.** Healthy controls (Control) are represented as a grey line and participants with unilateral vestibular hypofunction (vestibular) as a black line. The figure has a panel for static visuals and for dynamic visuals and the X axis represents the sound level (none, stationary white noise or real recorded sounds).

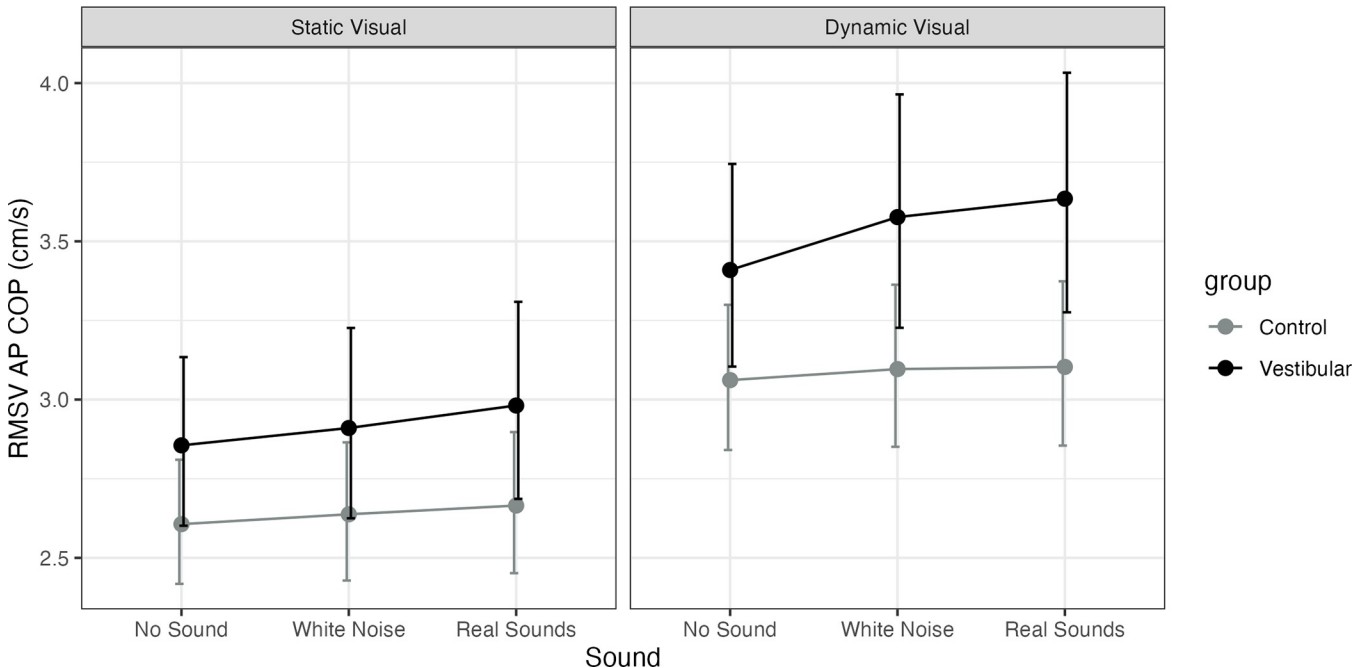

**Fig 3. Model estimated marginal means for anterior-posterior center-of-pressure (COP) Root Mean Square Velocity (RMSV, cm/s) and their 95% confidence intervals.** Healthy controls (Control) are represented as a grey line and participants with unilateral vestibular hypofunction (vestibular) as a black line. The figure has a panel for static visuals and for dynamic visuals and the X axis represents the sound level (none, stationary white noise or real recorded sounds).

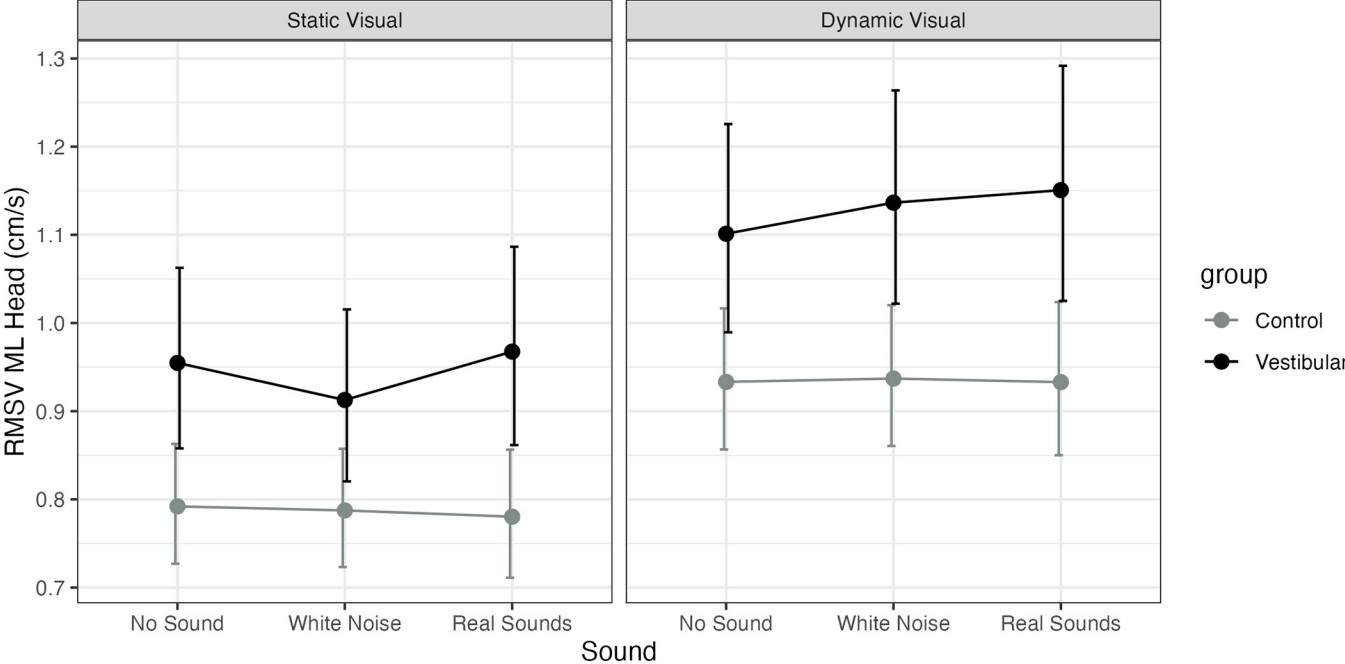

**Fig 4. Model estimated marginal means for head medio-lateral Root Mean Square Velocity (RMSV, cm/s) and their 95% confidence intervals.** Healthy controls (Control) are represented as a grey line and participants with unilateral vestibular hypofunction (vestibular) as a black line. The figure has a panel for static visuals and for dynamic visuals and the X axis represents the sound level (none, stationary white noise or real recorded sounds).

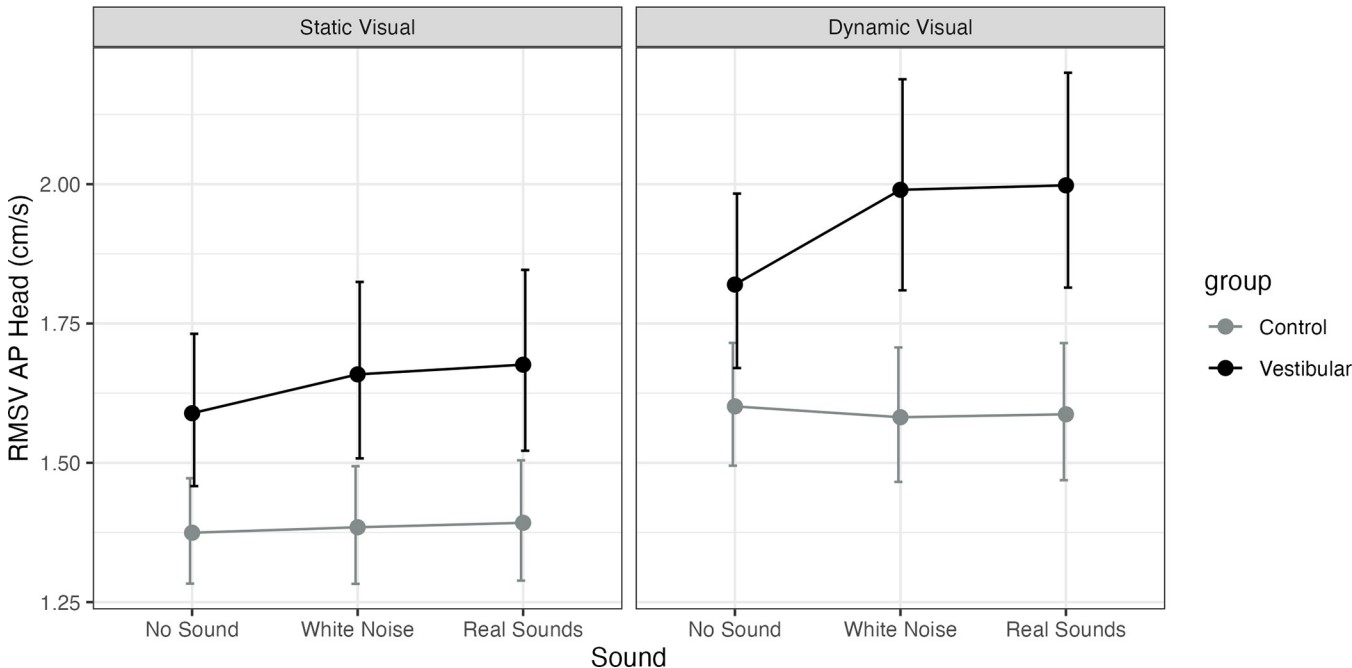

**Fig 5. Model estimated marginal means for head anterior-posterior Root Mean Square Velocity (RMSV, cm/s) and their 95% confidence intervals.**
Healthy controls (Control) are represented as a grey line and participants with unilateral vestibular hypofunction (vestibular) as a black line. The figure has a panel for static visuals and for dynamic visuals and the X axis represents the sound level (none, stationary white noise or real recorded sounds).

real P = 0.004, MED = 0.28; dynamic visual: no sound P = 0.023, MED = 0.22; white noise P<0.001, MED = 0.41; real P<0.001, MED = 0.41). A significant main effect of visual was observed for both groups across sound conditions (none, static, real, P<0.001) with no interactions. A significant increase in sway with real sounds vs. no sound was observed for the vestibular group on dynamic visuals (P = 0.003, MED = 0.18). See Fig 5 and S1 File.

**Pitch.** Adjusting for age the vestibular group was significantly higher than controls only on dynamic visuals with either sound (white noise P = 0.04, MED = 0.006; real P = 0.015, MED = 0.007). A significant main effect of visual was observed for both groups across sound conditions (none, static, real, 0.02≤P≤0.04) with no interactions. A significant increase in head pitch was observed with real vs. none (P = 0.001, MED = 0.006) and with white noise vs. none (P = 0.014, MED = 0.004) for the vestibular group on dynamic visuals. See Fig 6.

**Yaw.** Adjusting for age the vestibular group was significantly higher than controls only on dynamic visuals with real sounds (P = 0.036, MED = 0.004). A significant main effect of visual was observed for both groups only for real sounds (P = 0.047) with no interactions. A significant increase in sway with real sounds vs. no sound was observed for the vestibular group on dynamic visuals (P = 0.015, MED = 0.003). See Fig 7.

**Roll.** No differences between groups were observed. Main effect of visual for both groups only on dynamic sounds (P = 0.04) with no interactions. A significant increase in sway with real vs. none (P = 0.027, MED = 0.002) and with white noise vs. none (P = 0.017, MED = 0.002) was observed for the vestibular group on dynamic visuals. See Fig 8.

## Discussion

This study aimed to investigate how visual and auditory cues interact for postural control during a challenging standing balance task (standing on foam while wearing a virtual reality

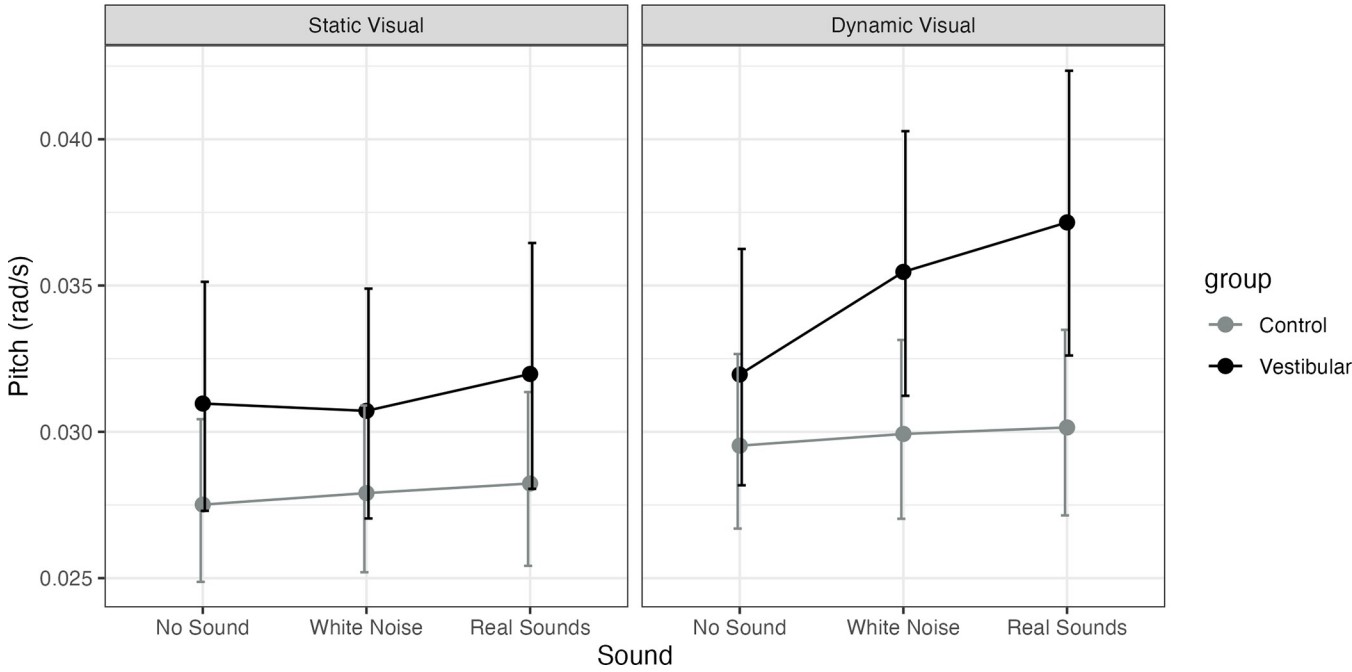

**Fig 6. Model estimated marginal means for head pitch Root Mean Square Velocity (RMSV, rad/s) and their 95% confidence intervals.** Healthy controls (Control) are represented as a grey line and participants with unilateral vestibular hypofunction (vestibular) as a black line. The figure has a panel for static visuals and for dynamic visuals and the X axis represents the sound level (none, stationary white noise or real recorded sounds).

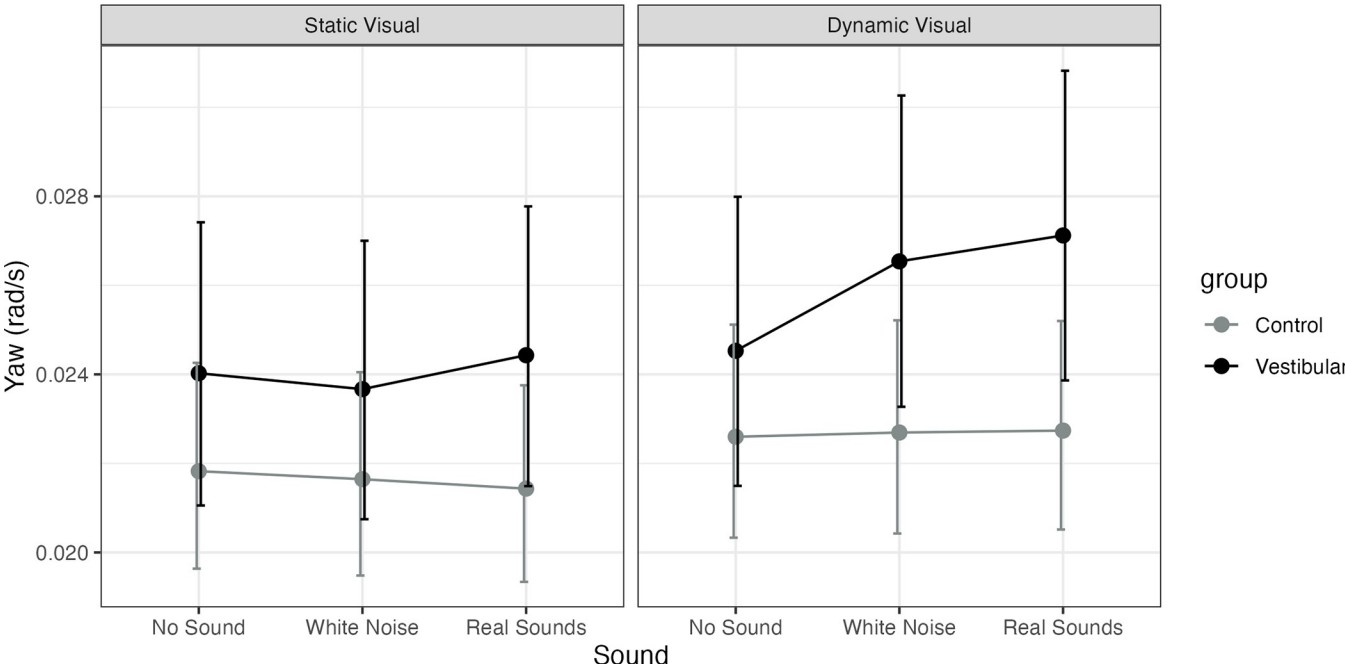

**Fig 7. Model estimated marginal means for head yaw Root Mean Square Velocity (RMSV, rad/s) and their 95% confidence intervals for healthy controls (Control) and participants with unilateral vestibular hypofunction (vestibular).** The figure has a panel for static visuals and for dynamic visuals and the X axis represents the sound level (none, stationary white noise or real recorded sounds).

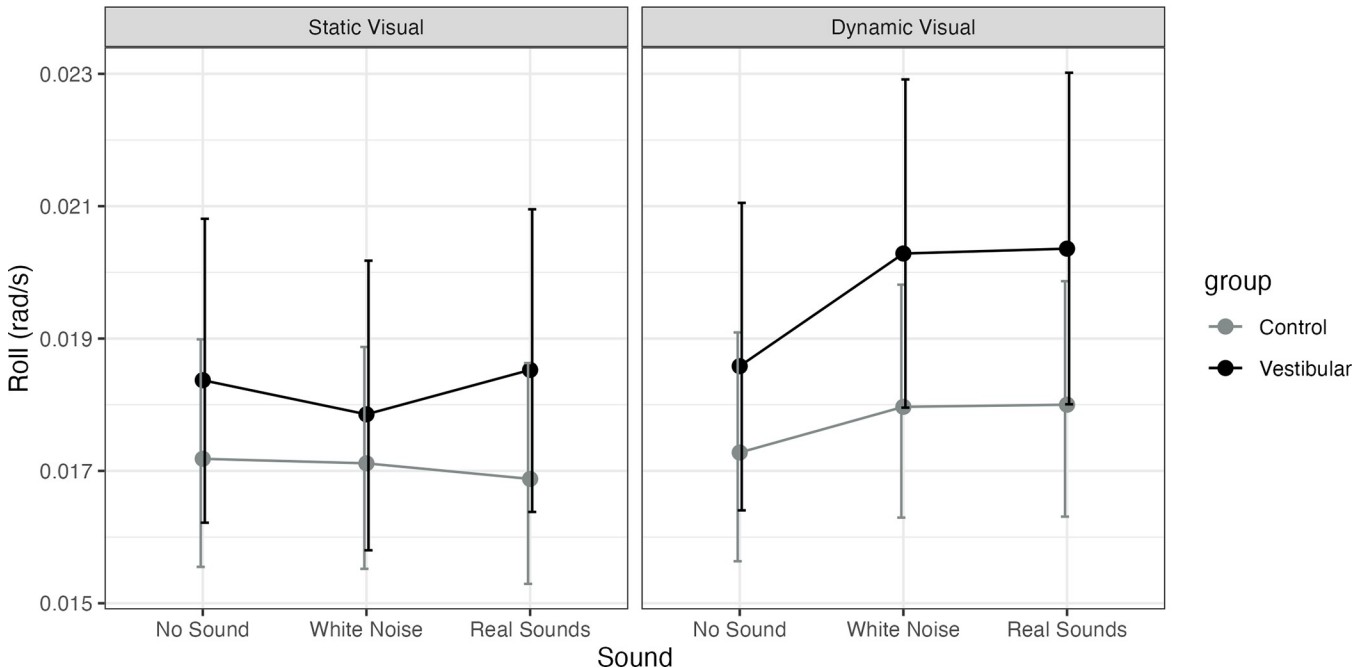

**Fig 8. Model estimated marginal means for head roll Root Mean Square Velocity (RMSV, rad/s) and their 95% confidence intervals for healthy controls (Control) and participants with unilateral vestibular hypofunction (vestibular).** The figure has a panel for static visuals and for dynamic visuals and the X axis represents the sound level (none, stationary white noise or real recorded sounds).

headset) with real-life visual context, broadband sounds (i.e., white noise) and real-recorded sounds. We chose to use spatialized real-recorded and broadband sounds because those have been shown to have more impact on balance than pure tones [8], and because those sounds have clear ecological validity. We studied people with a known sensory integration problem, i.e. unilateral peripheral vestibular hypofunction, and compared their balance performance to healthy controls. Both groups had normal hearing for their age confirmed by comprehensive behavioral audiometry. Our findings suggest that, in this specific experimental paradigm, balance responses of healthy adults aged 22–78 were not influenced by sounds. Differences observed across sound conditions were seen only for the vestibular group, only on the more challenging condition (i.e. dynamic visuals), and in most but not all directions (AP, Pitch, Yaw but not ML or Roll). In addition, the differences between groups were often magnified in the presence of either sound suggesting that- in this context (headphones, this specific visual environment, standing task)—white noise did not function as "auditory anchor" but rather as an additional perturbation. Taken together, these findings suggest that the role of sounds in postural control is highly context dependent: a healthy system can filter out unhelpful sounds to remain stable [17] whereas an impaired system perhaps cannot. Thus, sounds should be considered in balance evaluation and rehabilitation. Overall, our data suggest that in dynamic visual environments, sounds can be destabilizing for those with vestibular loss.

Our findings are consistent with the few studies that included individuals with sensory loss and showed that sounds impact postural control. Individuals with bilateral vestibular loss were destabilized by rotating cocktail party sounds [17]. Individuals with combined vestibular and hearing loss increased sway when sounds were blocked more so than controls did [13]. Vitkovic et al. observed increased sway in the attenuation of sounds (ear plugs or ambient room) that was larger in those with vestibular problems compared with healthy controls [13]. It is possible that a healthy multi-sensory system has multiple 'reserves' for balance control, and

does not require sounds as much as vision, somatosensory, and vestibular inputs for balance. In fact, our results underscore a conclusion of Vitkovic et al.: the importance of auditory cues for postural stability is minor compared with vision and somatosensory input [13]. Comparing the magnitude of response to the sensory perturbation, the largest effect size for the visual perturbation was in the AP direction and ranged from 0.43–0.46 cm/s for healthy controls and from 0.65–0.95 cm/s for vestibular hypofunction. The magnitude of change for the vestibular group between real sounds and no sounds was 0.22 cm/s. While it is methodologically challenging to create comparable stimuli, the auditory cues were matched to the visual cues (trains, footsteps etc.). The answer to our question "Would hearing a moving subway destabilize people similar to seeing the train move?" appears to be–not quite as much, but the additive effect of these stimuli created an additional layer of complexity that was harder for the patients with unilateral vestibular hypofunction.

The increase in sway with real contextual sounds may also be explained by the well-known phenomena of cognitive-motor interference: a decline in cognitive motor performance (in this case, reduced postural stability) while attempting to carry out two tasks simultaneously (i.e., dual-tasking) [30]. Even though participants in the current study were instructed to perform one task (maintaining their balance), they also needed to process auditory information which may have required attentional resources. Individuals with unilateral peripheral vestibular hypofunction require greater attentional resources than healthy controls to stabilize themselves due to their sensory loss [31]. Indeed, when comparing the performance of individuals with unilateral vestibular hypofunction to healthy controls on simple, inhibitory, and forced-choice reaction time (RT) cognitive tasks, even well-compensated patients had slower RTs compared to controls in standing and sitting [32]. Interestingly, several studies investigating cognitive-motor interference did not find significant differences between patients with vestibular hypofunction and controls [31–33]. Namely, both groups increased postural sway in a similar manner when asked to maintain balance while performing a cognitive task. In the current study, however, while both groups increased sway with dynamic vs. static visuals, only the vestibular group appeared to be influenced by the additional auditory perturbation. This may suggest that the primary challenge for the vestibular group was with sensory integration (reduced somatosensory information via the foam, dynamic visuals, added sounds and vestibular loss), i.e., inadequate weighting and re-weighting rather than cognitive load maxing out attentional capacity. This hypothesis needs to be investigated in future research comparing contextual sounds to auditory cognitive task in combination with dynamic visual load and their respective influence on balance.

Balance training is known to be task-specific, e.g., dynamic vs. static or surface type [34,35]. In this study, while the differences between groups were magnified in the presence of either sound in a dynamic visual environment, a main effect of sound was mostly observed for the real-recorded, salient sounds. Natural, real-world sounds provide a greater variety of binaural as well as monaural cues including static and moving features [36,37]. From the individual perspective, a person would have greater experience listening to natural sounds which may tap into emotional and memory aspects.

These findings have several implications to vestibular rehabilitation. First, sound perturbations should be included as part of both the assessment of balance and intervention programs. Ideally, these should be real sounds related to patients' typical environment and combined with salient and increasingly challenging visual cues. Both assessment and intervention can be operationalized in a clinical setting using portable virtual reality headsets. With respect to virtual reality testing of postural control, head sway appears to be sensitive to peripheral vestibular hypofunction [38], including to changes post-rehabilitation [39]. While we observed significant differences in pitch, and yaw, these differences were also observed in the AP

direction in a larger magnitude. That being said, in the specific task tested here we asked participants to look straight ahead. The importance of pitch and yaw, that can easily be obtained from the headset, increases during dynamic tasks [40,41].

### Study limitations

This study is limited by its cross sectional and laboratory nature. We attempted to create contextual visual and auditory cues, but these may not accurately reflect real life behavior. Research has shown greater impact of sounds on balance when using loudspeakers, but we chose to use headphones rather than loudspeakers for sound delivery because those are more applicable to clinical setting than an array of loudspeakers. We also tested standing balance only. The vestibular group was older, therefore all models adjusted for age. Given the challenging task (standing on foam), adjusting for age may have masked between-group differences in the ML direction.

### Conclusion

The addition of auditory stimuli, particularly contextually accurate sounds, to a challenging, standing balance task in an immersive virtual reality simulation increased postural and head sway in people with unilateral peripheral vestibular hypofunction but not in healthy controls. Between-groups differences were magnified in the presence of either sound suggesting that, in this context, white noise did not function as an anchor but rather as an additional perturbation. Clinicians should incorporate contextual sounds into their evaluation of balance in patients with sensory integration deficits, such as vestibular hypofunction and future research should develop and validate such instruments and programs.

### Supporting information

**S1 Video. An excerpt from the subway scene with dynamic visuals and real, recorded sounds.**
(MP4)

**S2 Video. An excerpt from the subway scene with static visuals and stationary, broadband sounds.**
(MP4)

**S1 File. Model estimate [95% Confidence Interval] for COP and Head AP and ML VRMS per group and sensory condition.**
(DOCX)

### Author Contributions

**Conceptualization:** Anat V. Lubetzky, Maura Cosetti, Daphna Harel, Agnieszka Roginska, Jennifer Kelly.

**Data curation:** Anat V. Lubetzky, Daphna Harel.

**Formal analysis:** Daphna Harel, Marlee Sherrod.

**Funding acquisition:** Anat V. Lubetzky, Maura Cosetti, Daphna Harel, Jennifer Kelly.

**Investigation:** Anat V. Lubetzky, Maura Cosetti, Marlee Sherrod, Zhu Wang, Agnieszka Roginska, Jennifer Kelly.

**Methodology:** Anat V. Lubetzky, Maura Cosetti, Daphna Harel, Zhu Wang, Agnieszka Roginska, Jennifer Kelly.

**Project administration:** Anat V. Lubetzky, Maura Cosetti, Jennifer Kelly.

**Resources:** Maura Cosetti, Zhu Wang, Agnieszka Roginska, Jennifer Kelly.

**Software:** Zhu Wang, Agnieszka Roginska.

**Supervision:** Anat V. Lubetzky, Maura Cosetti, Daphna Harel, Agnieszka Roginska, Jennifer Kelly.

**Validation:** Zhu Wang, Agnieszka Roginska.

**Visualization:** Daphna Harel, Marlee Sherrod, Zhu Wang.

**Writing – original draft:** Anat V. Lubetzky.

**Writing – review & editing:** Anat V. Lubetzky, Maura Cosetti, Daphna Harel, Marlee Sherrod, Zhu Wang, Agnieszka Roginska, Jennifer Kelly.

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
