## [Decision Letter · Decision Letter 0]

8 Dec 2024

PONE-D-24-39333Real Sounds Influence Postural Stability in People with Vestibular Loss but Not in Healthy ControlsPLOS ONE

Dear Dr. Lubetzky,

Thank you for submitting your manuscript to PLOS ONE. After careful consideration, we feel that it has merit but does not fully meet PLOS ONE’s publication criteria as it currently stands. Therefore, we invite you to submit a revised version of the manuscript that addresses the points raised during the review process.

Please submit your revised manuscript by Jan 22 2025 11:59PM. If you will need more time than this to complete your revisions, please reply to this message or contact the journal office at plosone@plos.org. Please include the following items when submitting your revised manuscript:A rebuttal letter that responds to each point raised by the academic editor and reviewer(s). You should upload this letter as a separate file labeled 'Response to Reviewers'.A marked-up copy of your manuscript that highlights changes made to the original version. You should upload this as a separate file labeled 'Revised Manuscript with Track Changes'.An unmarked version of your revised paper without tracked changes. You should upload this as a separate file labeled 'Manuscript'.

We look forward to receiving your revised manuscript.

Kind regards,

Renato S. Melo, PhD

Academic Editor

PLOS ONE

Journal Requirements:

Reviewers' comments:

Reviewer's Responses to Questions

**Comments to the Author**

1. Is the manuscript technically sound, and do the data support the conclusions?

Reviewer #1: Partly

Reviewer #2: Yes

2. Has the statistical analysis been performed appropriately and rigorously? 

Reviewer #1: Yes

Reviewer #2: Yes

3. Have the authors made all data underlying the findings in their manuscript fully available?

Reviewer #1: Yes

Reviewer #2: Yes

4. Is the manuscript presented in an intelligible fashion and written in standard English?

Reviewer #1: Yes

Reviewer #2: Yes

5. Review Comments to the Author

Reviewer #1: The manuscript compares the effects of auditory stimuli (no sound, white noise, and real-life environmental sounds—subway trains passing by in a station) on postural control in two visual conditions, static and dynamic (in an immersive contextual scene), simulating a real-life context in a virtual environment, in patients with unilateral vestibular loss and healthy controls. The testing was conducted on a foam surface with eyes open. The authors hypothesized that hearing real-life sounds would influence postural control in patients with vestibular deficits, as the vestibular system is a primary contributor to balance. They concluded that auditory stimuli, particularly contextually-relevant sounds, led to increased postural sway in vestibular patients but not in healthy controls during a standing balance task.

While the methodology appears sound overall, I have several major concerns that need to be addressed for the manuscript to be fully convincing.

Major Comments

- The authors did not adequately justify their findings based on known anatomical and physiological mechanisms. There are evidences on the integration of vestibular, auditory, and somatosensory inputs during postural control (Minino et al. 2023, Ann. Biomed. Eng., Review). The auditory system contributes to balance by providing spatial cues and better perception of the three-dimensional surrounding space. However, the manuscript lacks sufficient discussion on the links between auditory stimuli with vestibular function, particularly in the context of stance on an unstable surface (foam).

- More importantly, the study’s design introduces a dual-task paradigm, as participants are required to maintain postural control while processing auditory information. Cognitive involvement, such as attention and memory, plays a significant role in dual-task performance. Patients with vestibular hypofunction generally require greater cognitive resources to maintain postural stability due to their compromised sensory input. In dual-task situations, attention is divided between postural control and another concurrent task (in this case, auditory processing). This division of attention is known to impair postural control, especially in patients with vestibular deficits, who cannot fully allocate sufficient cognitive resources to both tasks. The authors did not address this aspect, which is crucial for interpreting their findings. Therefore, the manuscript should discuss the cognitive demands imposed by real-life sound stimuli and how this may exacerbate postural instability in vestibular patients.

Minor Comments:

- The participant information in Table 1 is typically presented in the Method section. I suggest removing the table and including the details in the text.

- The manuscript does not clarify whether there are significant statistical differences in demographic data between the patient and control groups. This should be included for transparency.

- These figures appear to repeat the data already reported in Table 3. To avoid redundancy, I recommend either removing the figures or presenting additional insights beyond the table data.

- In the figures, data are reported as means, with bars likely indicating standard deviations. Please confirm and clarify this in the figure legends.

- Significant differences in the data should be clearly marked with appropriate symbols in the figures.

- The age range of participants is mentioned for the first time in the Discussion section, and only for the healthy control group. This information should be included in the Method section and reported for both the patient and control groups.

Reviewer #2: A well desisgned study.

Minor comments

Replace ' Sample elegibilty criteria' by 'Elgibity criteria' or 'Inclusion and Exclusion Criteria' in the title of Table 1 and corresponding text.

Change Table 2 heading as Descriptive statistics

6. PLOS authors have the option to publish the peer review history of their article (what does this mean?). If published, this will include your full peer review and any attached files.

Reviewer #1: **Yes: **Mansoureh Adel Ghahraman

Reviewer #2: No

---

## [Author Response · Author response to Decision Letter 0]

18 Dec 2024

Summary of Revisions

Ref: PONE-D-24-39333

Title: Real Sounds Influence Postural Stability in People with Vestibular Loss but Not in Healthy Controls

Journal: PLOS ONE

Dear Prof. Melo fellow referees, 

We thank you for a thorough, detailed, constructive and very helpful review. We have made all of the corrections, and revised the manuscript according to your suggestions. We used track-changes font to highlight those changes made in the manuscript in response to the comments within the manuscript itself. In addition, below is a point-by-point list detailing how each of the major comments was addressed. 

Sincerely,

The Authors

Reviewer #1: The manuscript compares the effects of auditory stimuli (no sound, white noise, and real-life environmental sounds—subway trains passing by in a station) on postural control in two visual conditions, static and dynamic (in an immersive contextual scene), simulating a real-life context in a virtual environment, in patients with unilateral vestibular loss and healthy controls. The testing was conducted on a foam surface with eyes open. The authors hypothesized that hearing real-life sounds would influence postural control in patients with vestibular deficits, as the vestibular system is a primary contributor to balance. They concluded that auditory stimuli, particularly contextually-relevant sounds, led to increased postural sway in vestibular patients but not in healthy controls during a standing balance task.

While the methodology appears sound overall, I have several major concerns that need to be addressed for the manuscript to be fully convincing.

Major Comments

- The authors did not adequately justify their findings based on known anatomical and physiological mechanisms. There are evidences on the integration of vestibular, auditory, and somatosensory inputs during postural control (Minino et al. 2023, Ann. Biomed. Eng., Review). The auditory system contributes to balance by providing spatial cues and better perception of the three-dimensional surrounding space. However, the manuscript lacks sufficient discussion on the links between auditory stimuli with vestibular function, particularly in the context of stance on an unstable surface (foam).

Thank you for this comment. These connections are now added to the introduction (paragraph 1) to better set the stage for this scientific inquiry as well as to the discussion (paragraph 3).

- More importantly, the study’s design introduces a dual-task paradigm, as participants are required to maintain postural control while processing auditory information. Cognitive involvement, such as attention and memory, plays a significant role in dual-task performance. Patients with vestibular hypofunction generally require greater cognitive resources to maintain postural stability due to their compromised sensory input. In dual-task situations, attention is divided between postural control and another concurrent task (in this case, auditory processing). This division of attention is known to impair postural control, especially in patients with vestibular deficits, who cannot fully allocate sufficient cognitive resources to both tasks. The authors did not address this aspect, which is crucial for interpreting their findings. Therefore, the manuscript should discuss the cognitive demands imposed by real-life sound stimuli and how this may exacerbate postural instability in vestibular patients.

We added a discussion paragraph (paragraph 3) to address this point. Thank you!

Minor Comments:

- The participant information in Table 1 is typically presented in the Method section. I suggest removing the table and including the details in the text.

Done. 

- The manuscript does not clarify whether there are significant statistical differences in demographic data between the patient and control groups. This should be included for transparency.

This was added to Table 1. 

- These figures appear to repeat the data already reported in Table 3. To avoid redundancy, I recommend either removing the figures or presenting additional insights beyond the table data.

We believe that illustrating the results increases clarity for the readers. At the same time, any reader wishing to compare results, conduct a power analysis or report it in a review would need the exact estimates. That being said, given your suggestion, we moved Table 3 to an Appendix. 

- In the figures, data are reported as means, with bars likely indicating standard deviations. Please confirm and clarify this in the figure legends.

The data in the figures are model estimated marginal means and their 95% confidence intervals. This is now explained in the figure captions. 

- Significant differences in the data should be clearly marked with appropriate symbols in the figures.

Given the complexity of the models (main effect of group, visual load, and interactions separately for each auditory condition), we believe that including symbols in the figures would complicate the visualization whereas the P values reported in the text allow the reader to understand all the comparisons that we made per model. In addition, adding symbols to the figures would provide information regarding an arbitrary cutoff (0.05) whereas when the exact P values allow the readers to make their own judgement. We strongly believe that to be best statistical practice. 

- The age range of participants is mentioned for the first time in the Discussion section, and only for the healthy control group. This information should be included in the Method section and reported for both the patient and control groups.

Done. 

Reviewer #2: A well designed study.

Minor comments

Replace ' Sample eligibility criteria' by 'Eligibly criteria' or 'Inclusion and Exclusion Criteria' in the title of Table 1 and corresponding text.

The table was removed as per Reviewer 1’s request. The information is now presented in the text under the sub-hearing ‘eligibility criteria’.

Change Table 2 heading as Descriptive statistics

Done.

---

## [Decision Letter · Decision Letter 1]

8 Jan 2025

Real Sounds Influence Postural Stability in People with Vestibular Loss but Not in Healthy Controls

PONE-D-24-39333R1

Dear Dr. Lubetzky,

We’re pleased to inform you that your manuscript has been judged scientifically suitable for publication and will be formally accepted for publication once it meets all outstanding technical requirements.

Kind regards,

Renato S. Melo, PhD

Academic Editor

PLOS ONE

Additional Editor Comments (optional):

Reviewers' comments:

Reviewer's Responses to Questions

**Comments to the Author**

1. If the authors have adequately addressed your comments raised in a previous round of review and you feel that this manuscript is now acceptable for publication, you may indicate that here to bypass the “Comments to the Author” section, enter your conflict of interest statement in the “Confidential to Editor” section, and submit your "Accept" recommendation.

Reviewer #1: (No Response)

2. Is the manuscript technically sound, and do the data support the conclusions?

Reviewer #1: (No Response)

3. Has the statistical analysis been performed appropriately and rigorously? 

Reviewer #1: (No Response)

4. Have the authors made all data underlying the findings in their manuscript fully available?

Reviewer #1: (No Response)

5. Is the manuscript presented in an intelligible fashion and written in standard English?

Reviewer #1: (No Response)

6. Review Comments to the Author

Reviewer #1: (No Response)

7. PLOS authors have the option to publish the peer review history of their article (what does this mean?). If published, this will include your full peer review and any attached files.

Reviewer #1: **Yes: **Mansoureh Adel Ghahraman

---

## [Editor Report · Acceptance letter]

10 Jan 2025

PONE-D-24-39333R1 

PLOS ONE

Dear Dr. Lubetzky, 

I'm pleased to inform you that your manuscript has been deemed suitable for publication in PLOS ONE. Congratulations! Your manuscript is now being handed over to our production team.

Kind regards, 

on behalf of

Dr. Renato S. Melo 

Academic Editor

PLOS ONE